

# The relative contributions of scattering and viscoelasticity to the
# attenuation of S waves in Earth's mantle
Susini deSilva[1], Vernon F. Cormier[1]
[1]Department of Physics, University of Connecticut, 196 Auditorium Road, Storrs, CT 06269, USA
*Correspondence to*: Vernon F. Cormier (vernon.cormier@uconn.edu)
**Abstract.** The relative contributions of scattering and viscoelastic attenuation to the apparent attenuation of seismic
body waves are estimated from synthetic and observed S waves multiply reflected from Earth's surface and the core-
mantle boundary. The synthetic seismograms include the effects of viscoelasticity and scattering from small-scale
heterogeneity predicted from both global tomography and from thermodynamic models of mantle heterogeneity that
have been verified from amplitude coherence measurements of body waves observed at dense arrays. Assuming
thermodynamic models provide an estimate of the maximum plausible power of heterogeneity measured by elastic
velocity and density fluctuations, we predict a maximum scattering contribution of 43 % to the total measured
attenuation of mantle S waves having a dominant frequency of 0.05 Hz. The contributions of scattering in the upper
and lower mantle to the total apparent attenuation are estimated to be roughly equal.  The relative strength of the
coda surrounding observed ScSn waves from deep focus earthquakes is not consistent with a mantle having zero
intrinsic attenuation.
**1 Introduction**
Seismic tomography reveals a laterally heterogeneous velocity structure in the mantle. Constraining the locations
and dimensions of such elastic heterogeneities is critical to understanding the intricate details of the dynamic mixing
process of the mantle, which is closely tied to the plate tectonic evolution of the Earth. Large-scale (~ 1000 km)
heterogeneities are likely caused by the buoyancy differences that drive thermal-chemical convection. The effects of
thermal diffusion, however, limit small-scale (~ 1 to 100 km) heterogeneities to chemical variations. Small-scale
heterogeneities can scatter 0.1 to 1 Hz. body waves, transferring energy from body wave pulses observed at a
receiver to later time windows and receivers (Shearer, 2015). Mantle attenuation measured from P and S waves will
hence always be a summation of a scattering and an intrinsic viscoelastic attenuation. The viscoelastic dispersion of
dominantly intrinsic attenuation successfully explains the lower velocities of Earth models derived from low
frequency free oscillations observed in the millihertz band from those derived from 1 Hz body waves (Dziewonski
and Anderson, 1981).  Yet some extrapolations of the scale lengths and intensities of heterogeneity inferred from
high frequency body waves have suggested attenuation in the mantle may instead be dominated by scattering
(Ricard et al., 2014, Sato, 2019).



The apparent attenuation of multiple ScS waves is an excellent observable to untangle the relative contributions of
scattering and intrinsic attenuation. Many previous studies have used ScS and its reverberations within the mantle to
obtain path averaged values for the mantle attenuation. These attenuation measurements are usually represented in
terms of a quality factor (Q or $Q_{ScS}$ for ScS-based measurements). The estimates of these apparent attenuation
measurements include both the intrinsic or viscoelastic attenuation of the wave amplitude and the attenuation caused
by scattering effects. In this work we will consider the apparent attenuation ($\frac{1}{Q_{ScS}}$) to be the addition of intrinsic
attenuation ($\frac{1}{Q_{intr}}$) and scattering attenuation ($\frac{1}{Q_{scat}}$) for path averaged observations of SH waves reflected from the
free surface and core-mantle boundary. The intrinsic component accounts for the loss of energy due to friction and
heat loss as the wave propagates through the mantle with different viscous properties caused by the motion of
defects in the crystalline lattice structure of silicates or by the motion of melt at grain boundaries or in pores.
Intrinsic attenuation manifests itself in body waves by amplitude decay, pulse broadening, and velocity dispersion.
The scattering attenuation accounts for the energy loss that is scattered into different directions as elastic
heterogeneities are encountered along the path of a body wave. In addition to amplitude decay and pulse broadening
of the main phase, scattering generates increased levels of coda energy comprised of redistributed energy arriving
later than the main phase. Many past studies calculating the apparent attenuation of multiple ScS use spectral
amplitude ratios (Kovach and Anderson, 1964, Yoshida and Tsujiura, 1975, Sipkin and Jordan, 1980, Lay and
Wallace, 1983) and time domain amplitude ratios (Kanamori and Riviera, 2015) of adjacent ScS waveforms. An
alternative analysis technique seeks the attenuation operator that converts an $ScS_{n-1}$ waveform into an $ScS_n$
waveform (Jordan and Sipkin ,1977, Revenaugh and Jordan, 1989). Sipkin and Revenaugh (1994) concluded that a
frequency domain approach works better for $Q_{ScS}$ measurements, especially in continental regions that tend to have
lower shear Q values compared to oceanic regions. Lee et al. (2003) compared observations and numerical
simulations of coda envelope offsets before and after ScS synthesized with two-layer scattering models
superimposed on a PREM reference model to calculate the scattering contribution to total attenuation measurements.
They concluded that scattering loss dominates intrinsic loss in the lower mantle.
Our effort employs an estimate for a ScSn attenuation operator to evaluate the relative percentages of scattering and
intrinsic attenuation contributing to the apparent attenuation observed from simulated mantle heterogeneity models.
Observations of scattered body waves together with geodynamic modeling have established that heterogeneities of
scale lengths as small as 4 to 10 km with RMS (root mean square) velocity perturbations of 1 to 8 % can persist
throughout the mantle, even in the presence of constant convective stirring (Hedlin et al., 1997, Shearer and Earl,
2008, Kaneshima and Helffrich, 2010). Our investigation considers the effects of similar dimensions and
perturbation strengths for heterogeneity models. We also consider the effects of a maximum plausible depth-
dependent model of mantle heterogeneity power from thermodynamically constrained estimates of mantle chemistry
and phase. Such models predict significantly higher heterogeneity than the models of global tomography (Stixrude
and Lithgow-Bertelloni, 2007, Stixrude and Lithgow-Bertelloni, 2012). We have recently validated (Cormier et al.,
2019) a thermodynamic model of mantle heterogeneity by applying stochastic tomography (Zheng and Wu, 2008) to
the upper 1000 km of the mantle to invert for amplitude and phase fluctuations observed by the US transportable
array. Assessing the scattering attenuation induced by thermodynamic models, which predict heterogeneity to be





concentrated in mantle phase transition zones, can assist in quantifying mantle heterogeneity and testing for the
existence of additional phase transitions.

**2 Method**

**2.1 Models**

Apparent attenuations are measured from ScSn waveforms observed in synthetic seismograms for 4 different models
of mantle heterogeneity. All of these assume PREM as the one dimensional background velocity and density model,
with the PREM shear wave attenuations providing the purely intrinsic component of attenuation. Model 1 does not
perturb PREM with any lateral heterogeneities. Therefore, the apparent attenuation measured for this case will be
purely intrinsic. Model 2 (Fig.1) applies a depth-dependent shear velocity perturbation to the PREM mantle similar
to those determined from many seismic tomographic studies (Megnin and Romanowicz, 2000, Ritsema et al., 2004).
Model 3 (Fig. 2) applies a shear velocity perturbation to the PREM mantle similar to the predictions of
thermodynamic studies for the upper 1000 km of the mantle (Stixrude and Lithgow-Bertelloni, 2012). Model 4 (Fig.
3) is the same as Model 3 in the upper 1000 km of the mantle but includes an additional peak in heterogeneity power
in the lowermost mantle predicted from the effect of the post-perovskite phase transition. In Model 5, the intrinsic
attenuations are turned off while still applying the thermodynamic model of mantle heterogeneity to shear velocity
perturbations. Hence the synthetic seismograms for this model will exhibit purely scattering effects in any
attenuation measurement. In all models, heterogeneities are represented as stochastic random media with an
exponential autocorrelation having a corner scale equal to 10 km. In Models 2, 3, 4, and 5 we assume a relation
between density and shear velocity perturbations such that $d\ln\rho/\rho = 0.8\ d\ln Vs/Vs$. This value for density
perturbation in a mantle close to neutral buoyancy is relatively large, but is commonly assumed in studies of crustal
and upper mantle scattering based on Birch's law (Birch, 1952).

**2.2 Apparent attenuation measurements**

All simulations are performed by a numerical pseudospectral method in 2-D (Cormier, 2000), assuming an SH line
source at 500 km depth with a Gaussian-shaped source-time function having a half-width of 1.2 seconds. Wave
propagation uses a 2D staggered grid of radial step size 3.0 km and lateral step size 5.427 km, with time sampling
set to 0.025 seconds ensuring stability and negligible grid dispersion. Intrinsic attenuation, taken to be
approximately constant across a broad frequency band, is introduced by three memory functions using the methods
described by Robertson et al., (1994). Waveforms are computed at a great circle distance of 18° in order to avoid
contamination of ScSn phases with depth phases or other nearby arrivals. These are corrected for 3D geometric
spreading, and a line-to-point source conversion is made. For each of the 5 models a 2-parameter attenuation
operator (Eq. 1) is determined that converts the ScS waveform into an ScSScS waveform. Each attenuation operator





depends on $Q_{ScS}$ and the high frequency corner ($1/\tau_m$) of a relaxation spectrum, where attenuation is constant for 5
decades of frequency.
In the inversion procedure, the predicted ScSScS velocity waveform is generated by convolving the ScS waveform
with an attenuation operator corresponding to a peak attenuation $1/Q_{ScS}$ and a high frequency corner $1/\tau_m$. A least
squares norm is calculated (Eq. 2) for the difference between observed and predicted ScSScS velocity waveforms,
which are aligned by the arrival times of first maximum and normalized by the peak-to-trough amplitudes (Fig. 4).
A search over the two attenuation parameters is then performed to minimize an L2 norm difference to maximize a
Gaussian probability density constructed using the L2 norm difference (Cormier et al., 1998). Half-widths of the
probability density functions are used to infer errors.

An operator to convert an ScS waveform into an ScSScS waveform is defined in the frequency domain by

$$O(\omega, Q, \tau) = \exp(-i\omega[\int_{ScSScS} \frac{ds}{\hat{V}(\omega)} - \int_{ScS} \frac{ds}{\hat{V}(\omega)}]) \tag{1}$$
where
$$\hat{V}(\omega, Q, \tau) = \frac{\sqrt{1 + \dfrac{2}{\pi Q_{ScS}^{-1} \ln(\dfrac{-i\omega + {}^{1}/_{\tau_l}}{-i\omega + {}^{1}/_{\tau_m}})}}}{\sqrt{1 + \dfrac{2}{\pi Q_{ScS}^{-1} \ln(\dfrac{-i2\pi + {}^{1}/_{\tau_l}}{-i2\pi + {}^{1}/_{\tau_m}})}}}$$
and where
$\tau_l$ is the period of the low frequency corner in relaxation spectrum and $\frac{\tau_l}{\tau_m} = 10^5$
The least squares norm difference between observed and predicted waveforms is calculated from

$$L2N(Obs, Pred) = \sqrt{\sum_t \frac{(Amp_{obs}(t) - Amp_{pred}(t))^2}{\sigma^2}} \tag{2}$$
where $\sigma$ is a $\frac{noise}{signal}$ measurement from a 100 second time window preceding the ScSScS observation.

Our goal was to simply estimate an apparent attenuation parameter $Q_{ScS}$ for the whole of the mantle when the effects
of scattering are included rather than to seek a best fitting depth and frequency dependent attenuation model. Our
estimates for the high frequency corner parameter $1/\tau_m$, were within the range bounded by estimates for the upper
and lower mantle found in the study by Choy and Cormier (1986).

**3. Results**



We found MODEL 1, which has pure intrinsic attenuation and no small-scale heterogeneity, to have an apparent
attenuation value of 0.004167 corresponding to a $Q_{ScS}$ = 240. This estimated $Q_{ScS}$ value differs by only 2.2 % from
the theoretical estimate of the depth averaged $Q_{ScS}$ obtained for PREM with the relation $Q_{ScS} = (\int_{x\_ScSScS} dt -$
$\int_{x\_ScS} dt)/(\int_{x\_ScSScS} dt/Q_s(x) - \int_{x\_ScS} dt /Q_s(x))$. Here $x\_ScSScS$ and $x\_ScS$ denote points along the path of
ScSScS and ScS respectively, Qs(x) denote the Qs values at those points read from 1D PREM. This result verifies
the accuracy of the waveform L2 norm method for estimating $Q_{ScS.}$

With MODEL 2, which has a conventional tomographic estimate of mantle heterogeneity, we find that the apparent
attenuation is increased to 0.005 ($Q_{ScS}$ decreased to 200). Together with the knowledge of the purely intrinsic
contribution ($\frac{1}{Q_{intr}}$) calculated in MODEL 1, the scattering component of attenuation ($\frac{1}{Q_{scat}}$) in MODEL 2 is
estimated to be 0.000833. Hence the scattering caused by small-scale (~ 10 km) heterogeneities with a dVs/Vs depth
profile similar to S20RTS (Ritsema et al., 2004), would account for 16.7 % of the measured ScS apparent
attenuation. MODEL 3, which has a higher amount of heterogeneity due to increased Vs perturbations associated
with predicted lateral variations in phase changes in the upper mantle, results in a higher apparent attenuation of
0.005747 ($Q_{ScS}$ = 174). MODEL 4, which includes additional heterogeneity predicted for the effects of a post-
perovskite phase transition results in an even higher apparent attenuation of 0.007100 ($Q_{ScS}$ = 140). We calculate
that the scattering attenuation in the lower mantle (below 1000 km) and upper mantle (above 1000 km) of MODEL
4 to be 0.0014 and 0.0016 with their percent contributions to the total apparent attenuation being 19.6 % and 22.4 %
respectively. The overall scattering attenuation of MODEL 4 is 0.002933 with the scattering component accounting
for 41.3 % of the measured ScS total apparent attenuation.
Finally, in MODEL 5 the intrinsic attenuation in the mantle is turned off while applying the mantle heterogeneity of
MODEL 4. The apparent attenuation (now purely due to scattering) is measured to be 0.0029 ($Q_{ScS}$ = 340). This high
Q value lies towards the upper bound of regional estimates (~ 360) of $Q_{ScS}$ (Nakanishi, 1979, Sipkin & Revenaugh
1994, Gomer & Okal, 2003). It is also found that apparent attenuation measurements of MODEL 5 and MODEL 1
add up to be exactly equal to MODEL 4, validating the attenuation estimation method in conjunction with the
assumption of $\frac{1}{Q_{apparent}} = \frac{1}{Q_{intr+scat}} = \frac{1}{Q_{intr}} - \frac{1}{Q_{scat}}$.

Figure 6 compares the levels of scattered coda energy arriving in the vicinity (~ ± 150 s) of the ScSScS main arrival
generated by different models of mantle heterogeneity models to the synthetic ScSScS predicted by MODEL 1
having no scattering. Observing the envelopes of squared velocity for MODEL 2 versus MODEL 4, it is apparent
that the levels of energy arriving in the coda and before the main phase significantly increase and the ScSScS pulse
width increases due to the presence of increased small-scale heterogeneity in the regions associated with mantle
phase changes. It also is important to recognize that intrinsic attenuation can affect the ratio of coda energy to the
main pulse. The results for MODEL 5, which omits intrinsic attenuation, demonstrate the importance of intrinsic
attenuation for the coda as well as the direct phases. In this case the coda, unaffected by intrinsic attenuation,
approaches the amplitude of the direct ScSScS phase.




## 4. Discussion


### 4.1 Comparison with regional variations



Regional variations measured for $Q_{ScS}$ generally fall in the range of 140 – 360 (Nakanishi, 1979, Sipkin &
Revenaugh, 1994, Gomer & Okal 2003). Variations on this order are confirmed when we apply our inversion
method to two example multiple ScS observations observed from deep focus earthquakes (Fig. 7). We obtain $Q_{ScS}$ =
153 for an earthquake beneath Papua New Guinea region observed at a station located at Charters Towers in
Australia, and $Q_{ScS}$ = 200 for an earthquake beneath the eastern China-Russia border region observed at a station
located at Yakutsk in eastern Siberia. In Fig. 8 we overlay synthetic seismograms computed from several of our
models to determine of how scattering in combination with intrinsic attenuation can affect the relative amplitudes of
the direct ScSScS phase and its coda. The heterogeneity power of MODEL 2 inferred from global tomography is too
weak to match the excitation of coda relative to ScSScS in both our data examples. MODEL 4, having PREM
attenuation and heterogeneity predicted for a thermodynamic model of the mantle, best matches the relative coda
and direct phase excitations for both events. The match can be improved by either a small decrease in intrinsic
attenuation or a small increase in heterogeneity power for the eastern China-Russia border region to Yakutsk. ScSn
paths from both earthquakes traverse a region of the mantle on the back-arc side of dipping slabs, a southwest
dipping slab toward the Australian craton in the case of the New Guinea event (Tregoning and Gorbatov, 2004), and
a western dipping Kuril-Kamchatka slab (Koulakov et al., 2011) toward the Siberian craton in the case of the eastern
China-Russia border event. The multiple ScSn paths for the eastern China-Russia border event are more slab
parallel and distant from the descending slab and more strongly sample the cratonic upper mantle compared to the
New Guinea event. Hence, it is likely that the intrinsic attenuation of PREM overestimates the effects of mantle
attenuation on ScSn's. Finally, a comparison of observations with the prediction of Model 5, having no intrinsic
attenuation, over-predicts coda excitation relative to ScSScS for both events. This confirms that some intrinsic
attenuation in the mantle is necessary to dampen the coda generated by the most extreme plausible suggestions of
heterogeneity power.

### 4.2 Upper and lower mantle scattering and intrinsic attenuation


Strong depth dependence of mantle attenuation, both intrinsic and scattering, has long been documented. Intrinsic
attenuation has been found to be relatively low in the mid and deep mantle compared to the upper mantle. Evidence
of some scattering in the mid and deep mantle has been confirmed in studies of PKIKP precursors in the 120° to
140° great circle range (e.g., Hedlin et al., 1997), including strong regional and depth variations that may be
consistent with the effects of either remnant subducted oceanic crust or with a peak in heterogeneity power
associated with a post-perovskite phase change. From a study of S and ScS coda, Lee et al. (2003) estimated that
scattering attenuation dominates intrinsic attenuation in the lower mantle, reporting their results in terms of the



scattering coefficients for a two-layered model of mantle heterogeneity. The scattering coefficients g are related to
scattering attenuation by g = omega/($Q_{scat}$ Vs).  Our results for MODEL 3 and MODEL 4 show that seismic albedo,
the ratio of scattering loss to total attenuation, below 1000 km depth in the mantle is 30 % while above 1000 km it is
27 %. This is assuming the PREM average intrinsic shear Q of 225 and 312 for the two depth regions. Hence, we do
not observe the scattering to dominate over intrinsic effects in either lower or upper mantle, although regional
exceptions can be expected.  Additionally, considering the estimated scattering attenuations for MODEL 3 and
MODEL 4, we can deduce the scattering coefficients to be $6.25 \times 10^{-5}$ km$^{-1}$ for the mantle below 1000 km and 1.256
$\times 10^{-4}$ km$^{-1}$ for mantle above 1000 km in MODEL 4. These scattering coefficients, calculated for a dominant
frequency of 0.05 Hz, are comparable to the low frequency estimates of Lee et al. (2003). This result implies a
relatively lower scattering coefficient (i.e. slightly lower scattering attenuation) in the lower mantle compared to the
upper mantle in MODEL 4, which agrees with the Lee et al. estimates of scattering coefficients.

**4.3 Origins of heterogeneity and scale length anisotropy**

In suggesting that scattering attenuation may dominate intrinsic attenuation throughout the mantle Ricard et al.
(2014) considered the effects of heterogeneity distributed primarily in the form of horizontal layers based on
geodynamic numerical experiments that predict folding and horizontal stretching of chemical heterogeneity (e.g.,
Manga, 1996) whose origin primarily originates from the convective cycling of oceanic crust.  The attenuative
effects of horizontally layered structure have been well known since the classic paper by O'Doherty and Anstey
(1971) and are simply calculated. In this paper, we have instead considered the effects of scale lengths predicted by
thermodynamic models in which variations in temperature and chemistry dictate the stability of silicate mineral
phases.  These variations in temperature and chemistry can also be connected to the convective cycling of oceanic
crust, but instead predict that peaks in heterogeneity power will be concentrated near phase transitions. Such models
have not yet fully considered the effects of mechanical mixing on the anisotropy of scale lengths within these
relatively narrow regions of depth. Nonetheless, thermodynamic models, when verified by observations of scattering
effects that supplement tomographic imaging, may at least provide a more reliable estimate of the upper bound to
velocity and density fluctuations in the mantle.  Experiments similar to ours may be extended to include the effects
of anisotropy of scale lengths.  Our results indicate that some intrinsic attenuation will always be required to explain
the attenuation of body waves, regardless of the state of isotropy of scale lengths.

**5. Conclusions**

An inversion algorithm for apparent mantle attenuation based on L2 norm differences between observed and
predicted ScSScS velocity waveforms has been verified by inversion of synthetic seismograms and applied to
estimate the relative contributions of intrinsic and scattering attenuation to the total apparent attenuation.
Thermodynamic models of mantle heterogeneity predict significantly higher heterogeneity power than the
predictions from global tomography, and a correspondingly higher relative contribution to apparent attenuation



measured from body waves. Taking the depth-dependent heterogeneity power of thermodynamic models of mantle
heterogeneity as the maximum plausible heterogeneity we estimate that scattering may explain up to 41.3 % of
apparent mantle attenuation with up to 3 % RMS shear velocity perturbations concentrated near mantle phase
transitions and 1 % everywhere else. We estimate the scattering contribution to the apparent attenuation from
heterogeneity in the upper and lower mantle to be roughly equal in global averages, but regional variations between
upper and lower mantle scattering contributions are likely. These estimates agree well with the excitation of coda
surrounding ScSn waves observed from deep focus earthquakes. These codas can only be matched by the existence
of both intrinsic and scattering attenuation.

**Data Availability.** The data set of SH component synthetic seismograms can be found at
https://doi.org/10.5281/zenodo.3460694 (Desilva and Cormier, 2019).

**Acknowledgements.** This work was supported by grants EAR 14-46509 from the National Science Foundation.

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

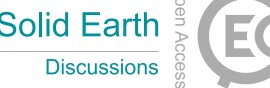




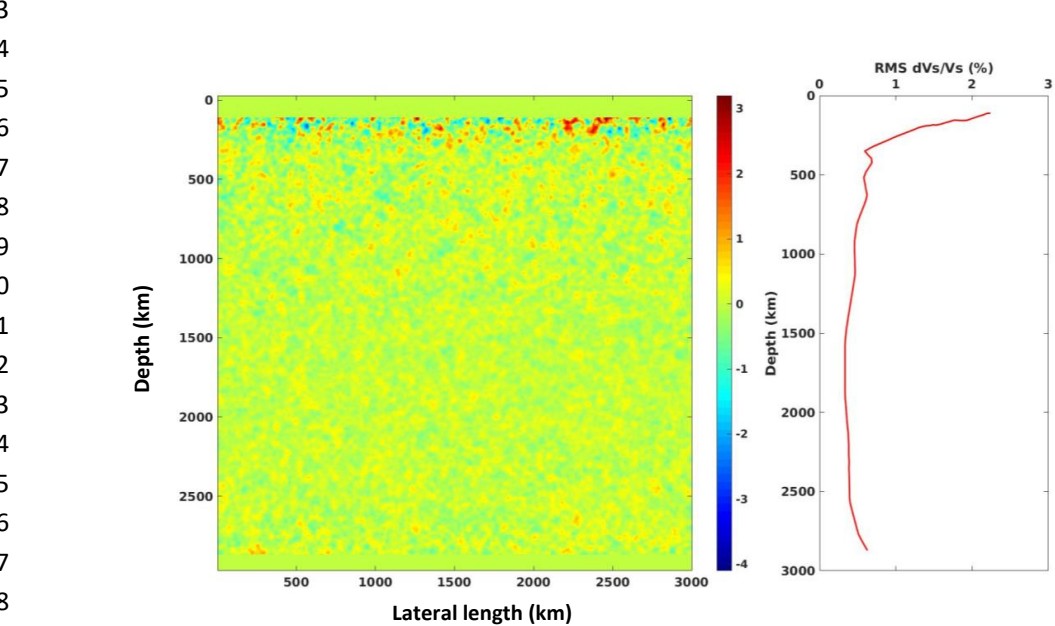

**Figure 1: Right: depth dependent RMS shear velocity perturbation profile applied in Model 2. This is extracted from**
**S20RTS. Left: 2D representation of the same depth dependent profile. Heterogeneous media is for an exponential**
**autocorrelation (corner scale a = 10 km) function. Note the increase in heterogeneity power near the top and bottom of**
**the mantle.**




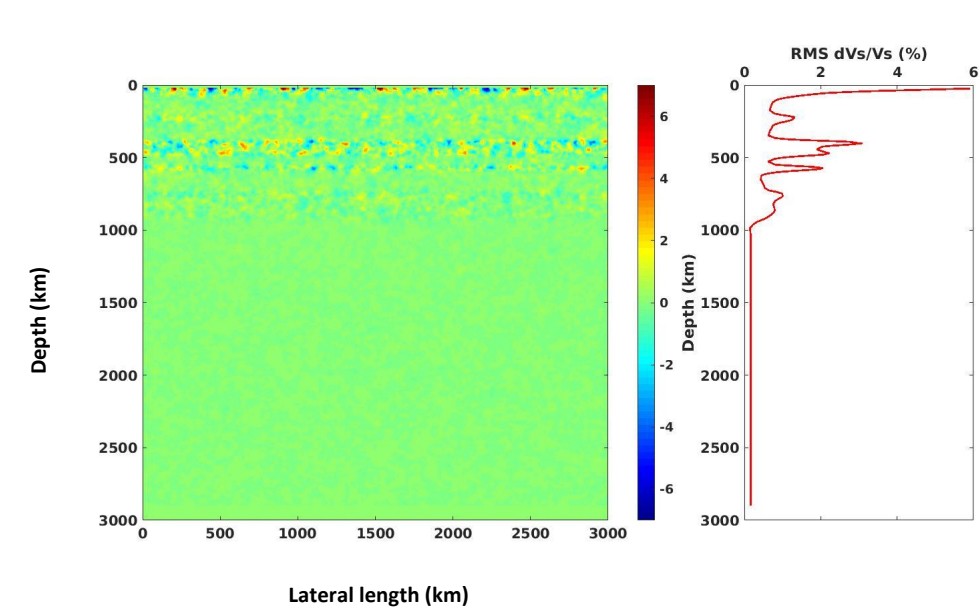

**Figure 2: Right: Depth dependent RMS shear velocity perturbation profile applied in Model 3 vs. perturbation values**
**from crust to 1000 km depth is extracted from the stochastic tomography result of Cormier et al. (2019). Left: 2D**
**representation of the same depth dependent profile. Compared to Model 2 note the additional peaks in heterogeneity**
**power associated with phase transitions in the upper mantle.**



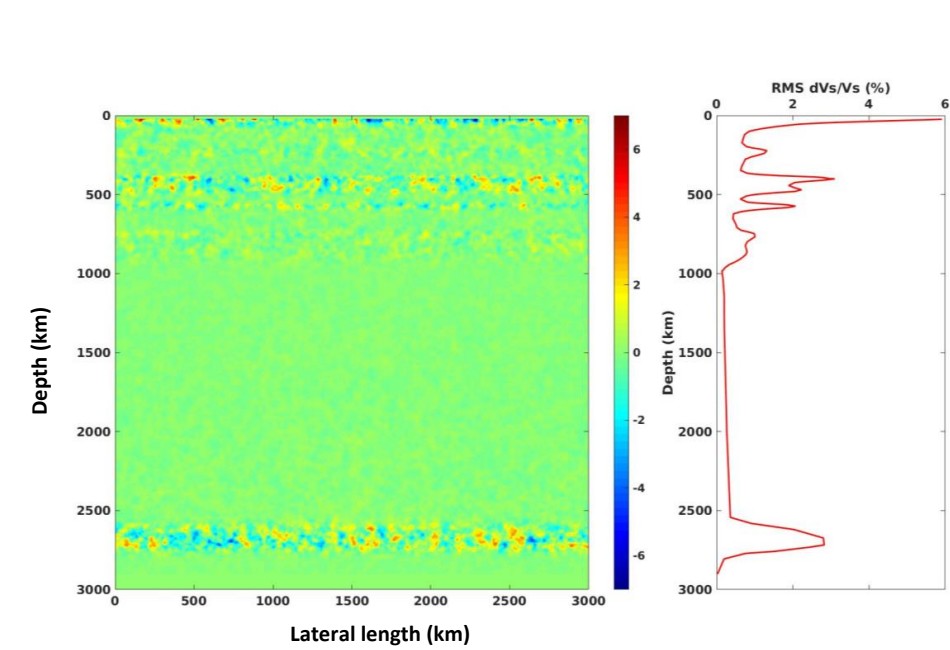

**Figure 3: Right: Depth dependent RMS shear velocity perturbation profile applied in Model 4 vs. perturbation values from crust to 1000 km depth is extracted from the stochastic tomography result of Cormier et al. (2019). Compared to Model 3 an additional peak is added near the core mantle boundary to incorporate the increased lower mantle associated with the post-perovskite phase change (Stixrude & Lithgow-Bertelloni, 2012) Left: 2D representation of the same depth dependent profile.**




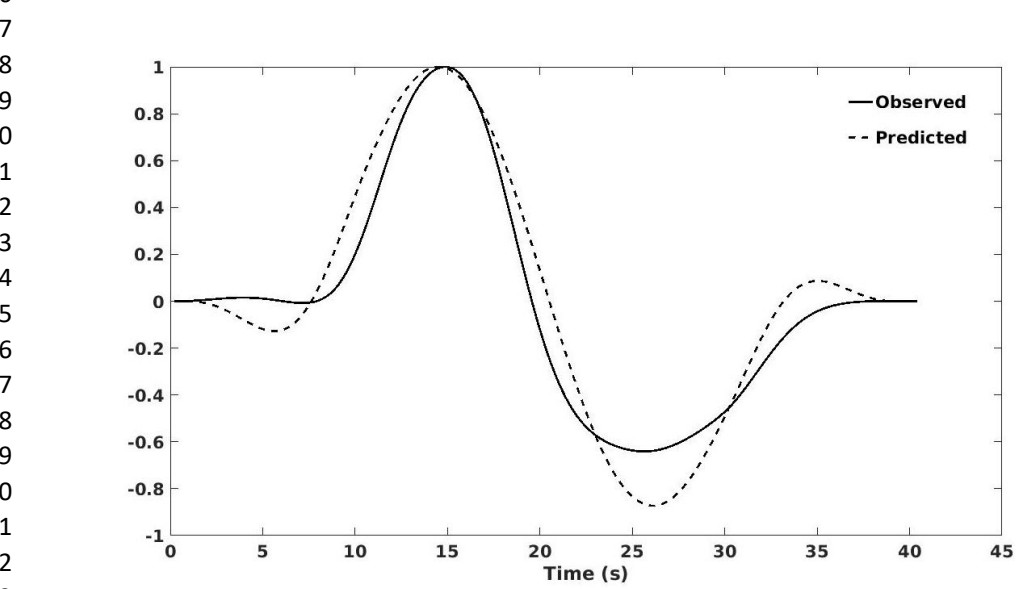

**Figure 4: Observed and predicted ScSScS velocity waveform aligned by the arrival time of first extremum and normalized by the peak to trough amplitude. The least squares norm difference between these two waveforms is obtained using a summation of amplitude differences over time.**






*(a)*

*(b)*

*(c)*

*(d)*

*(e)*

**Figure 5: Gaussian probability density function constructed with the least squares norm difference between predictions and simulated observations for (a) MODEL 1, (b) MODEL 2, (c) MODEL 3, (d) MODEL 4 and (e) MODEL 5.**






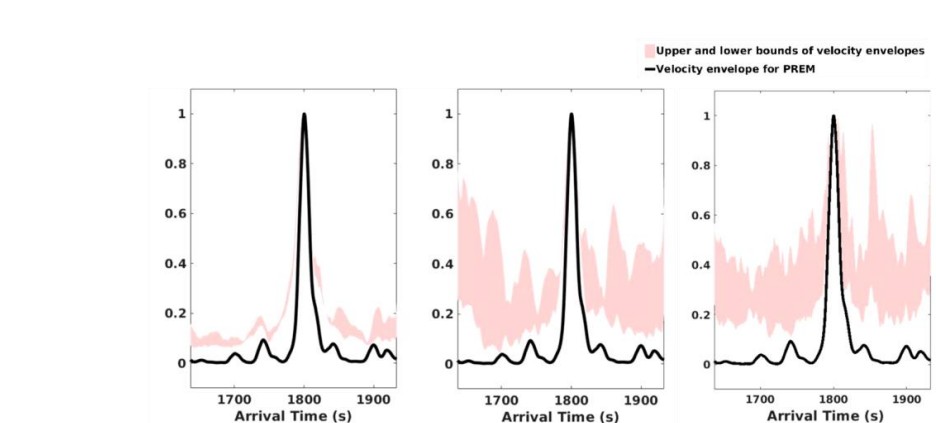

**Figure 6: Upper-lower bounds of coda envelopes (shaded area) calculated from 5 random heterogeneity realizations of**
**each MODEL 2 (left), MODEL 4 (middle) and MODEL 5 (right), compared to PREM (black line).**




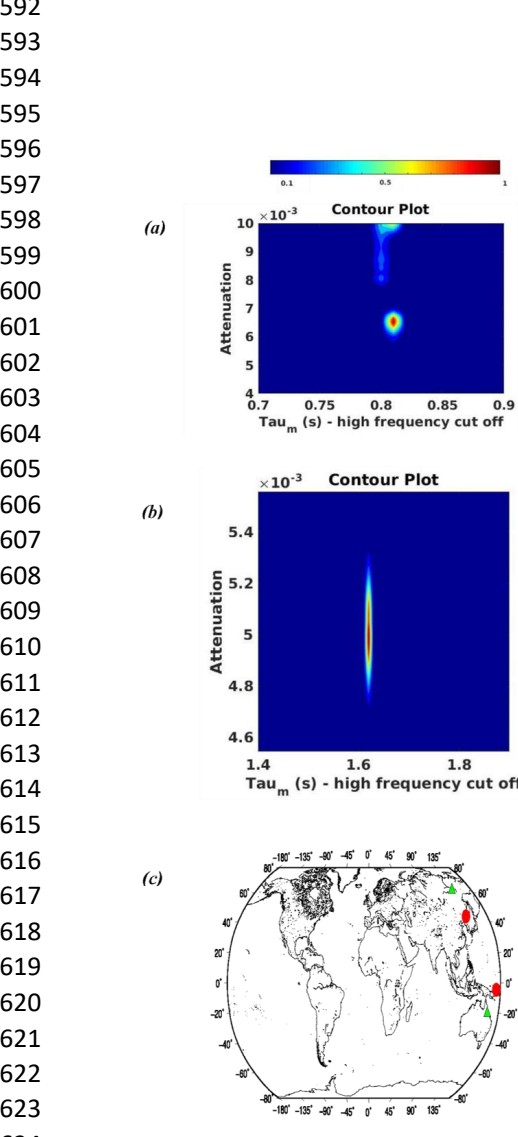

**Figure 7: Contour plots of probability density functions obtained with multiple ScS observations in two regions. Event (circles) and IU station (triangles) locations for the two regions described below are shown in panel (c).**

    **(a)** **Mantle beneath Papua New Guinea region : Observations are recorded by station CTAO (146.25° E, 20.08° S) for a 490 km deep, mw 6.6 event (154.88° E, 45.43° S) which occurred on May 02 1998, 13:34:28 UTC. Event-station distance is 17.6°.**

    **(b)** **Mantle beneath Eastern China-Russia border region : Observations are recorded by station YAK (129.68° E, 62.03° N)  for a 568 km deep, mw 7.3 event (130.66° E, 43.76° N) which occurred on June 28 2002, 17:19:30 UTC. Event-station distance is 18.3°.**





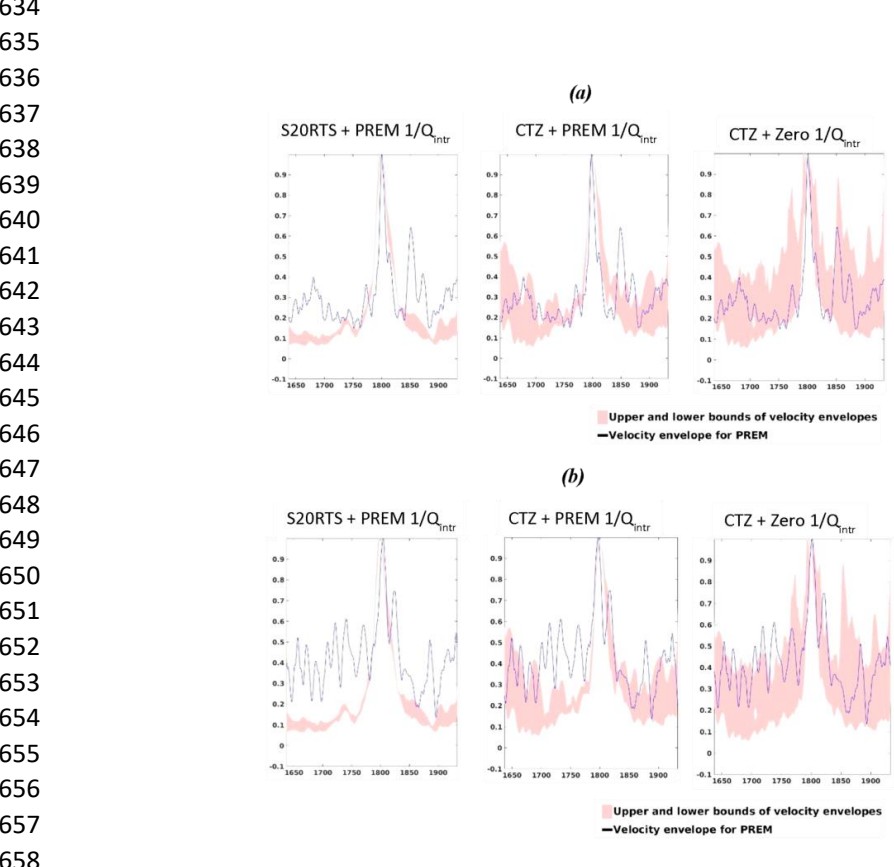

**Figure 8: Upper-lower bounds of coda envelopes (shaded area) calculated from 5 random heterogeneity realizations of**
**each MODEL 2 (left), MODEL 4 (middle) and MODEL 5 (right), compared to the squared velocity envelopes of data**
**traces (solid blue lines) from (a) Papua New Guinea data and (b) Eastern China-Russia border region.**




|  | $Q_{ScS} \pm \delta Q_{ScS}$ | $\tau_m \pm \delta\tau_m$ (sec) |
|---|---|---|
| MODEL 1 | 0.004167 ± 0.00028 | 3.800 ± 0.004 |
| MODEL 2 | 0.005000 ± 0.00034 | 3.790 ± 0.004 |
| MODEL 3 | 0.005747 ±0.00066 | 4.600 ±0.010 |
| MODEL 4 | 0.007100 ± 0.0005 | 3.630 ± 0.007 |
| MODEL 5 | 0.002900 ± 0.0003 | 1.980 ± 0.005 |


**Table 1 : Apparent attenuation parameters and their errors estimated for the five simulated models using probability density functions shown on Fig. 5..**







| | $Q_{ScS}$ | *Scattering Attenuation* *Apparent Attenuation* | *Intrinsic Attenuation* *Apparent Attenuation* |
|---|---|---|---|
| MODEL 1 (PREM) | 240 | | 100 % |
| MODEL 2 (Tomographic dVs/Vs model (exponential ACF, a = 10km) + PREM ) | 200 | 16.7 % | 83.3 % |
| MODEL 3 (Thermodynamic dVs/Vs model for UM only (exponential ACF, a= 10 km) + PREM ) | 174 | 27.5% | 72.5% |
| MODEL4 (Thermodynamic dVs/Vs model for both UM and LM (exponential ACF, a = 10 km) + PREM) | 140 | 41.3 % | 58.7 % |
| MODEL 5 (Thermodynamic heterogeneity + no intrinsic attenuation + PREM velocities and densities) | 340 | 100 % | |

**Table 2 : Estimated relative contributions to apparent 1/$Q_{ScS}$.**