# Peer review of "The relative contributions of scattering and viscoelasticity to the attenuation of S waves in Earth's mantle"

_Solid Earth, 2019_

## Referee Comment (RC1) · Ian Jackson (Referee) · 31 Oct 2019

General comments

This analysis by de Silva and Cormier is a welcome addition to discussion of the relative contributions of scattering and viscoelastic relaxation to the apparent attenuation of seismic shear waves in the Earth's mantle.

The heterogeneity responsible for the scattering of seismic waves is introduced by consideration of random media with a characteristic scale length of 10 km. ScS and ScSScS waveforms are computed for wave propagation in 2D through such media

with correction for 3D geometric spreading. An attenuation operator, relating ScS and ScSScS waveforms via a constant-Q absorption band, is determined for each of 5 models that combine viscoelastic relaxation and heterogeneity responsible for scattering in different proportions. This analysis thus constrains the relative contributions of viscoelastic relaxation and scattering to the total attenuation. The relative amplitudes of direct arrivals and related coda are also simulated.

Two principal conclusions emerge from this analysis: (i) scattering alone cannot account for the observations, but (ii) the coda observations require more intense scattering than predicted from the heterogeneity associated with tomographic wave speed models.

Specific comments

In my opinion, there are several aspects of the analysis that require more elaboration and discussion as follows. Firstly, does the 2D analysis of the wave propagation bias the estimated intensity of scattering by ignoring scattering into and out of the plane of the calculation? Secondly, to what degree are the results of this analysis influenced by the assumption of constant Q within the absorption band, rather than the mild frequency dependence (Q $\sim$ f1/3) consistently revealed by laboratory studies? Thirdly, how was the thermodynamic model of mantle heterogeneity derived? In particular, what range of variability of chemical composition and temperature was allowed? Fourthly, what is the explanation for the conclusion that the heterogeneity from the tomographic wavespeed model is insufficient to explain the amplitude of the ScS coda? Does this potentially reflect the fact that spatial smoothing tends to mean that the amplitudes of wavespeed anomalies are underestimated?

Ian Jackson 31 October 2019

---

## Referee Comment (RC2) · Anonymous Referee #2 · 3 Nov 2019

The manuscript presents synthetic seismogram analyses of ScS multiple attenuation varying the importance of intrinsic attenuation and scattering in five different model scenarios. A simple comparison with the range of ScS Q values from a few prior studies and analysis of two earthquakes in this study is used to estimate the approximate balance between scattering and intrinsic attenuation in the upper and lower mantle. The modeling aspect of the study is well-conducted and the five scenarios provide a new and instructive perspective on the tradeoffs between scattering and intrinsic attenuation. The connection of the modeling results to inferences about Earth's mantle via comparison with observational results is much weaker on account of the choice to ignore the wealth of relevant and easily accessible seismic data in modern community

archives. Consequently, I am cautious about the value of the interpretations regarding the balance of scattering and intrinsic attenuation in the real rather than synthetic model mantle. The observational component of the manuscript should be substantially expanded to use global data from many sources and a large number of receivers as the available data resources have advanced greatly beyond those used in most of the references. Comparing a more statistically significant set of waveform analyses to the modeling results would be a powerful approach for evaluating the relative influences of scattering and intrinsic attenuation.

Given the quality of the modeling component I would suggest focusing on that in this manuscript and refraining from insights into actual mantle properties rather than just model implications. Or, with much more observational analysis a compelling observational component could be added to this study.

---

## Short Comment (SC1) · 8 Nov 2019

Reviewer: The manuscript presents synthetic seismogram analyses of ScS multiple attenuation varying the importance of intrinsic attenuation and scattering in five different model scenarios. A simple comparison with the range of ScS Q values from a few prior studies and analysis of two earthquakes in this study is used to estimate the approximate balance between scattering and intrinsic attenuation in the upper and lower mantle. The modeling aspect of the study is well-conducted and the five scenarios provide a new and instructive perspective on the tradeoffs between scattering and intrinsic attenuation. The connection of the modeling results to inferences about

[Figure]

Earth's mantle via comparison with observational results is much weaker on account of the choice to ignore the wealth of relevant and easily accessible seismic data in modern community archives. Consequently, I am cautious about the value of the interpretations regarding the balance of scattering and intrinsic attenuation in the real rather than synthetic model mantle. The observational component of the manuscript should be substantially expanded to use global data from many sources and a large number of receivers as the available data resources have advanced greatly beyond those used in most of the references. Comparing a more statistically significant set of waveform analyses to the modeling results would be a powerful approach for evaluating the relative influences of scattering and intrinsic attenuation. Given the quality of the modeling component I would suggest focusing on that in this manuscript and refraining from insights into actual mantle properties rather than just model implications. Or, with much more observational analysis a compelling observational component could be added to this study.

Author Reply: In the interest of considering clear ScS and ScSScS phases uninterfered by depth phases and other arrivals (eg: S, SS, sS), as pointed out in section 2.2, authors prefer the use of deep events and observations in $10 - 30$ degree distance range. While authors agree that an analysis of the full observational data set satisfying said conditions would be quite valuable to better constrain predictions regarding the real mantle, the main objective of the study is to set up a well-defined modelling method and illustrate how this perspective can be applied on observational data. Hence the current quantitative predictions of scattering vs. intrinsic attenuation contributions are restricted to the mantle regions sampled by the considered previous studies and presented earthquake data.

In the attached Fig. 1 we illustrate the currently available event clusters with preferable depths and appropriate moment magnitudes ($> 6$ Mw) from all catalogs of IRIS DMC 1970-01-01 – 2019-11-07, and in Fig. 2 the best distance range to observe uninterfered clear signals of ScS/ScSScS is highlighted ($< 30$ degrees), for the use of

future observational studies applying the discussed method to resolve mantle atten-uation characteristics in a global scale. We note that source radiation pattern must be kept in mind when searching for high SNR multiple ScS on transverse component seismograms.

Approximate total numbers of land-based stations (permanent and temporary experi-ments) available around each regional cluster of suitable events currently available in IRIS DMC are listed below. Tonga-Kermadec region – up to 50, Papua New Guinea region – up to 100, Banda/ Java sea region – up to 200, Philippine island region – up to 200, Japan/ eastern China region – up to 50, Peru/ Chile region – up to 100.

[Figure]

**Fig. 1.**

[Figure]

**Fig. 2.**

---

## Author Comment (AC2) · 8 Nov 2019

Reviewer: The connection of the modeling results to inferences about Earth's mantle via comparison with observational results is much weaker on account of the choice to ignore the wealth of relevant and easily accessible seismic data in modern community archives. Consequently, I am cautious about the value of the interpretations regarding the balance of scattering and intrinsic attenuation in the real rather than synthetic model mantle. The observational component of the manuscript should be substantially expanded to use global data from many sources and a large number of receivers as the available data resources have advanced greatly beyond those used in most of the

references. Comparing a more statistically significant set of waveform analyses to thermodynamic modeling results would be a powerful approach for evaluating the relative influences of scattering and intrinsic attenuation. Given the quality of the modeling component I would suggest focusing on that in this manuscript and refraining from insights into actual mantle properties rather than just model implications. Or, with much more observational analysis a compelling observational component could be added to this study.

Addressing the reviewer's comment on use of dense modern waveform data, we point out that the analysis and conclusions of our paper are supported by a heterogeneity model that was determined from many 100's of waveforms observed from deep focus earthquake in 3 regions recorded by sensors in the USArray of Earthscope. Fig. 1 attached from the publication that resulted from that study plots the deep focus earthquakes and stations we used in that analysis. The peaks in heterogeneity power determined from that study closely match those predicted in the theromodynamic mantle models of Stixrude and Lithgow-Bertelloni. The only feature that was not directly determined from the model of maximal heterogeneity was the additional peak we added for heterogeneity concentrated near a post-perovskite phase transition that appears in the thermodynamic models. For examining effects of heterogeneity on S waves we assumed a scaling of dlnVs/dlnVp = 2. Fig. 2 shows a plot of our upper mantle heterogeneity P velocity fluctuations (red) with our estimated error bars is compared with the predictions of the thermodynamic model (blue).

The close correspondence of our estimated heterogeneity and that of the thermodynamic model with an assumed scaling between P and S velocity fluctuations, combined with several publications that have proposed that a significant fraction of mantle attenuation is due to scattering (e.g., Yicard et al. , EPSL, 2014, doi: 10.1016/j.epsl.2013.12.008), primarily motivated the current study.

The supporting section discussing the analysis of attenuation inverted from 2 representative deep focus earthquakes is important for demonstrating that coda levels and

waveform complexity as well as the shape of the initial pulse are also important for constraining the effects of heterogeneity. We chose events that we felt best represented the span of attenuation values determined in earlier works published by Sipkin, Jordan, Revanaugh, Okal, and others. We acknowledge the reviewer's point that a much larger volume of data is available for reexamining the mantle attenuation from ScSn observations. A voluminous data sample would enable stacking to more quantitatively characterize coda shapes. Our goal in this paper, however, was to simply demonstrate some maximal bounds on the contribution of scattering by choosing some representative events with quite different levels of apparent scattering in their ScSn codas. Our wish was to simply pursue some obvious consequences of our paper on stochastic tomography bearing on suggestions that the scattering attenuation exceeds intrinsic attenuation of body waves. We do not feel that a new global study of ScSn waveforms is necessary to support the conclusions of our paper, but we do agree that such a study is certainly ripe for a revisit.

———————————————

[Figure]

**Fig. 1.**

[Figure]

Fig. 2.

---

## Author Response (AR1)

Reply to Ian Jackson's review:

**Reviewer**: Does the 2D analysis of the wave propagation bias the estimated intensity of scattering by ignoring scattering into and out of the plane of the calculation?

*Authors' response*: Few tests of 2-D vs 3-D scattering effects exist. Wu and Irving (GJI, 2017, doi: 10.1093/gji/ggx047), who compared 3-D to 2.5D numerical simulations, show an example test. Their 2.5D simulations do not remove the energy of out-of-plane scattering, but they did not find significant differences between the simulations for smoothed PKiKP coda.

At any given velocity and density perturbation, however, 3-D scattering in principle should remove more energy from the direct arriving pulse than 2-D scattering in a plane containing the source, receiver, and center of the earth. Thus the assumption of 2-D scattering at a given perturbation level will potentially overestimate the true perturbation level needed to produce that apparent attenuation. Any overestimate of the perturbation level just reinforces our conclusion that intrinsic attenuation is also needed to explain the observed apparent attenuation combined with coda levels.

*Authors' changes in manuscript*: We added lines addressing this comment in lines 109-111.

*Reviewer:* To what degree are the results of this analysis influenced by the assumption of constant Q within the absorption band, rather than the mild frequency dependence (Q f 1/3) consistently revealed by laboratory studies?

*Author's response*: Resolving the frequency dependence of intrinsic attenuation from seismic data is a notoriously difficult problem, complicated by depth dependence, and the need to compare observations over a very broad frequency band. Using observations of free-oscillations and low frequency surface waves (0.001 to 0.005 Hz), a study by Lekic et al. (EPSL, 2009, doi:10.1016/j.epsl.2009.03.030) found a power of 0.3 frequency dependence of Qs, diminishing to 0 as frequency increased. A body wave study by Choy and Cormier (JGR, 1986, doi: 10.1029/JB091iB07p07326) found small or no frequency dependence of attenuation in the upper and lowermost mantle but attenuation decreasing with frequency as a power of -1 above a corner frequency in the mid-mantle. The frequency band of our observations and simulations, however, is too narrow (0.01 to 0.25 Hz) to observe a difference between the effects of a power of 0 or - 0.3 for the frequency dependence of attenuation (1/Q). Our study also did not consider the complication of depth dependent changes in the shape of the relaxation spectrum, which would require both multiple S and ScS observations over a series of ranges

*Authors' changes in manuscript:* We added lines addressing this comment in lines 136-140

*Reviewer*: How was the thermodynamic model of mantle heterogeneity derived? In particular, what range of variability of chemical composition and temperature was allowed?

*Authors' response*: Except for a peak in heterogeneity power associated with a post-perovskite phase change in the lowermost mantle our test "maximum plausible" heterogeneity model was derived from a study of P wave coherence beneath the USArray (Cormier et al., Commun. Comp. Phys., preprint, doi: 10.4208/cicp.OA-2018-0079) , assuming dlnVs/dlnVp = 2. The peaks in the heterogeneity model inferred from P wave coherence closely coincide with predictions from thermodynamic models by Stixrude and Lithgow-Bertelloni, with which we were initially surprised.   These thermodynamic models considered a range of mantle compositions and mixing scenarios. Details are given in several of their papers.  Models of mantle compositions included both pyrolite and depleted MORB mantle, assumed both mechanical mixing and equilibrium assemblages, and considered variations in potential temperature between 1000 to 2000 deg K. Most of the differences between the models were their predictions for 1-D averages of mantle seismic velocities. There were not large differences between models for the size and position of predicted peaks of heterogeneity power at different depths, which are most important to seismic scattering.

*Auhtors' changes in manuscript*: We added lines addressing this comment in lines 65-79.

*Reviewer:* What is the explanation for the conclusion that the heterogeneity from the tomographic wavespeed model is insufficient to explain the amplitude of the ScS coda? Does this potentially reflect the fact that spatial smoothing tends to mean that the amplitudes of wavespeed anomalies are underestimated?

*Authors' response*: The heterogeneity power in these models is too weak to explain the observed ScS coda power even when we assumed a white spectrum between the scale lengths (>200 km) they can resolve and the scale length corresponding to the smallest scale (25 km) that will produce significant scattering the frequency pass band we observed and modeled.  Yes, images from global tomography underestimate wavespeed anomalies by smoothing fluctuations in travel time picks. These fluctuations are due to a combination of picking errors and the effects of unresolvable small-scale structure observed over paths limited in spatial density.  Regularizing parameters in tomographic inversion damp these fluctuations.  To explain multi-pathing that has been observed in body waveforms some studies have multiplied the velocity perturbations in large-scale structures imaged by tomograms by factors up to 2 to explain the observed waveform complexity (e.g., Romanowicz et al., EPSL 233, 137-153, 2005).

*Authors changes in manuscript:* We added lines and a reference addressing this comment in lines 200-204.

Reply to anonymous reviewer:

**Reviewer:** The connection of the modeling results to inferences about Earth's mantle via comparison with observational results is much weaker on account of the choice to ignore the wealth of relevant and easily accessible seismic data in modern community archives. Consequently, I am cautious about the value of the interpretations regarding the balance of scattering and intrinsic attenuation in the real rather than synthetic model mantle. The observational component of the manuscript should be substantially expanded to use global data from many sources and a large number of receivers as the available data resources have advanced greatly beyond those used in most of the references. Comparing a more statistically significant set of waveform analyses to thermodynamic modeling results would be a powerful approach for evaluating the relative influences of scattering and intrinsic attenuation. Given the quality of the modeling component I would suggest focusing on that in this manuscript and refraining from insights into actual mantle properties rather than just model implications. Or, with much more observational analysis a compelling observational component could be added to this study.

*Authors response:* Addressing the reviewer's comment on use of dense modern waveform data, we point out that the analysis and conclusions of our paper are supported by a heterogeneity model that was determined from many 100's of waveforms observed from deep focus earthquake in 3 regions recorded by sensors in the USArray of Earthscope. The thermodynamic models are cited simply to point out their agreement with the heterogeneity spectrum we obtained from P wave coherence together with an assumed scaling between P velocity fluctuations and S velocity and density fluctuations. We clarified the connections between our heterogeneity model and the thermodynamic models in revised lines 66 -79. See also some supporting figures uploaded in our earlier reply submitted during the discussion period.

The section discussing the analysis of attenuation inverted from 2 representative deep focus earthquakes is important for demonstrating that coda levels and waveform complexity as well as the shape of the initial pulse are also important for constraining the effects of heterogeneity.  We chose events that we felt represented the span of attenuation values determined in earlier works published by Sipkin, Jordan, Revanaugh, Okal, and others.  We acknowledge the reviewer's point that a much larger volume of data is available for reexamining the mantle attenuation from ScSn observations.  A voluminous data sample would enable stacking to more quantitatively characterize coda shapes.  Our goal in this paper, however, was to simply demonstrate some maximal bounds on the contribution of scattering by choosing some representative events with quite different levels of apparent scattering in their ScSn codas.

*Authors' changes in manuscript:* We clarified this in revised lines 184-191 and provide a supplementary figures showing possible data available for a more complete analys and also provide a synthetic SH record section for a deep focus earthquake to illustrate where ScS and ScSScS are observable and well separated from other phases.

[revised manuscript text omitted]